# The Predictors of Driving Cessation among Older Drivers in Korea

**DOI:** 10.3390/ijerph17197206

**Published:** 2020-10-01

**Authors:** SeolHwa Moon, Kyongok Park

**Affiliations:** 1College of Nursing, Hanyang University, 222 Wangsimni-ro, Sungdong-gu, Seoul 04763, Korea; seora@hanyang.ac.kr; 2Department of Nursing, Gangneung-Wonju National University, 150 Namwon-ro, Heungeop-myeon, Wonju-si, Gangwon-do 26403, Korea

**Keywords:** driving cessation, environment, mobility limitation, older drivers, quality of life, traffic accident

## Abstract

*Background*: As the elderly population and the number of older drivers grow, public safety concerns about traffic accidents involving older drivers are increasing. Approaches to reduce traffic accidents involving older drivers without limiting their mobility are needed. This study aimed to investigate the driving cessation (DC) rate among older Korean adults and predictors of DC based on the comprehensive mobility framework. *Method*: In this cross-sectional study, data from 2970 to 10,062 older adults over 65 years old from the 2017 National Survey of Elderly People were analyzed in April 2020. Multivariate logistic regression analyses were conducted to identify the predictors of DC. *Results*: Residential area, an environmental factor, was a strong predictor of DC (Odds Ratio (OR) 2.21, 95% Confidential Interval (CI) 1.86–2.62). Older drivers living in an area with a metro system were 2.21 more likely to stop driving than those living in an area without a metro system. Other demographic, financial, psychosocial, physical, and cognitive variables also predicted DC. *Conclusion*: Environmental factors were strong predictors of older adults’ DC. Therefore, political and environmental support, such as the provision of accessible public transportation, is essential to increase the DC rate among older adults to increase public safety without decreasing their mobility.

## 1. Introduction

Mobility is defined as the ability to move oneself, including by walking, driving or using transportation from one’s home to the community [1]. A high level of mobility gives older adults more opportunities to participate in social activity and eventually decreases isolation from society [2]. The maintenance of mobility is known to be fundamental to active aging, as it allows older adults to continue to lead dynamic and independent lives [3]. By contrast, impaired mobility is known to be associated with falling, decreasing of independence, being admitted to facilities, and even death because mobility is closely linked to health status and quality of life [1]. Factors that restrict older adults’ mobility, such as driving cessation (DC) to due advanced age, have negative impacts on their health status and reduce their quality of life [1,4]. 

Driving is a convenient transportation method that can be used whenever and wherever people want to move. Driving is one of the instrumental activities of daily living [5] and seems to be a simple skill for people. However, driving requires not only cognitive skills, such as attention, memory, problem solving, and information processing, but also physical ability [6]. Therefore, appropriate functioning of the brain, upper and lower extremities, and ears and eyes is required to drive a car [7]. As the elderly population grows, the number of older drivers also increases. For the majority of older adults, cars are the primary form of transportation and provide autonomy, flexibility and independence [8]. In America, there were approximately 28 million licensed drivers aged 70 and older in 2017. The number of licensed drivers aged 70 and older increased by 58% between 1997 and 2017, and the proportion of licensed drivers aged 70 and older will continue to increase in America in the near future [9]. In Korea, there were 611,714 licensed drivers over 65 years old in 2018 [10]. According to statistics, approximately 80% of males and 50% of females aged 70 and over will be licensed holders in 2030 in the UK [11]. The increasing number of older drivers as well as age-related functional change and disease have contributed to the increasing incidence of car accidents among the elderly population [4,12]. In Korea, there were a total of 10,155 car accidents involving older drivers in 2008, but this number increased to 30,012 in 2018 [10]. When a traffic accident involves an older driver, the possibility of a serious or death-causing accident increases compared to that when no older driver is involved [13]. Even if a car accident is mild, an older driver can simultaneously be the person who caused the accident and the victim of the accident because of their physical weakness, low resilience, and high rate of experienced complications [14,15]. Because of the risk of car accidents caused by older adults, most countries have established strict systems to screen older adults’ ability to drive. In Japan, drivers aged 70 and older have to be screened for cognitive function and take a driving lesson and driving test before renewing their driver’s licenses [12]. In Korea, it is recommended that drivers aged 65 and older take an optional lesson on safe driving, while for drivers aged 75 and older, the lesson is mandatory. As an incentive, the insurance rate is discounted for older drivers who take the safe driving lesson in Korea [16].

When we consider public safety related to car accidents caused by older adult driving, we should also consider the individual mobility of older adults because driving is also a primary form of transportation for this population as it is for other age groups [17]. If driving by older driver was restricted for public safety, their mobility would also be restricted by imposed DC and their independence and quality of life would be affected by restricted mobility. There is no evidence showing advanced age as the common reason to stop driving. Moreover, it is debatable whether older drivers are more unsafe than young drivers. One study reported no significant difference in the rate of accidents between age groups [18]. Other studies reported that older drivers drove more safely than younger drivers, and another study indicated that older drivers with medical conditions relevant to driving were safer drivers than younger drivers with medical conditions relevant to driving [19,20]. However, some studies reported a higher rate of accidents among older drivers than among drivers from other age groups [10,12]. Therefore, we need to consider the reasonable criteria that will prohibit high risk driving besides age, because driving restrictions due to advanced age that do not consider alternative transportation methods could restrict the mobility of older adults. A study also suggested that older adults’ independence and quality of life related to driving should be equally considered alongside public safety concerning car accidents caused by older drivers [6].

A comprehensive model of mobility in older adults identifies five factors that affect mobility: cognitive, psychosocial, physical, environmental, and financial factors. It is widely known that cognitive, psychosocial, and physical factors, such as having dementia, depression or a fear of falling, or visual impairment, influence mobility [1]. This comprehensive model also explains that environmental or financial factors, such as where individuals live and their financial status, are also determinants of mobility [1,21]. Access to public transportation is a good example of an environmental factor. Korea has metro systems in six cities, one of which is the Seoul metro system. Korean older adults prefer using the metro over the bus because the metro is clean and comfortable, a proper temperature and indoor environment are maintained in the metro and free metro rides are provided as social support [22]. As an environmental factor, the free metro ride program is a form of political support of the mobility of older adults in Korea. The program was started in 1980 by the Welfare of the Aged Act No. 25 and offers free rides for older adults aged over 65 years old, children under six years old, and disabled individuals. All six cities with metro systems implement the free ride program for older adults [23]. In Korea, 434,863 older adults aged over 65 used subways without paying a fare in 2018 [24]. Programs that make the use public transportation convenient for older adults can increase the number of older adults who use public transportation instead of driving a car and will eventually contribute to decreasing the car accident rate and increasing the mobility of older adults.

Therefore, the current study aims to examine the proportion of older adults who have stopped driving, investigate the relationships between DC and related factors (e.g., cognitive, psychosocial, physical, environmental, and financial factors), and identify the predictors of DC in community-residing older adults based on a comprehensive mobility framework [1]. 

## 2. Materials and Methods

### 2.1. Participants

We analyzed data from the 2017 National Survey of Elderly People in this cross-sectional study. The data from the 2017 National Survey of Elderly People were collected by stratified two-stage cluster sampling based on the number of older people included in the 2015 Census in Korea [25]. After the approval of the Korea Institute for Health and Social Affairs (KIHASA), the raw data were provided without any personally identifiable information. The Institutional Review Board of the Gangneung-Wonju National University approved the waiver of consent for this study (GWNUIRB-R2020-17). The data of 2970 of 10,062 older adults aged over 65 years were analyzed in this study after the data of 7092 adults who answered that they had never driven in their lives were excluded.

### 2.2. Measurements

#### 2.2.1. Driving Status

The driving status was classified among the older adults who had driving experience during their lifetime. The data for participants who had never driven in their lives were excluded. Driving status was measured by the item “Are you driving currently?”. Participants were required to choose from the following options: “Currently driving” and “Have driven but not currently driving”. A response of “Currently driving” was classified as “Driving”, and a response of “Have driven but not currently driving” was classified as “DC”.

#### 2.2.2. Demographic Factors

Age, gender, and level of education were measured as demographic factors related to driving.

#### 2.2.3. Financial Factor

Perceived financial status was measured as the financial factor. The response options for perceived financial status were “Good” and “Poor”.

#### 2.2.4. Psychosocial Factors

The psychosocial factors included items regarding depression and perceived social activity. Depression was measured by the Korean version of the Geriatric Depression Scale Short Form (SGDS). The SGDS consists of 15 items, which are scored on a scale ranging from 0 to 15. Higher scores indicate more depressive symptoms [26]. Social activity was assessed with the following question: “Have you participated in social groups in the past year?”. The possible answers were “Yes” and “No”.

#### 2.2.5. Environmental Factor

For the environmental factor, whether the residential area where participants lived had a metro system was assessed. Residential area was classified as either “Living in an area with a metro system” or “Living in an area without a metro system”.

#### 2.2.6. Physical Factors

The physical factors included perceived health, disease diagnosis (e.g., cataract, glaucoma or stroke), and lower and upper extremity function. Visual and hearing discomfort were classified as answers to the following questions: “Is there any discomfort of vision or hearing in daily life?”. A response to the question of “not uncomfortable” was classified as “no discomfort”, and responses of “uncomfortable” and “very uncomfortable” were classified as “with discomfort”. To assess lower extremity function, participants were asked whether they could perform a variety of tasks, such as sitting and standing up from a chair and climbing 10 stairs without rest, five times. To assess upper extremity function, participants were asked whether they could perform a task such as lifting or moving an object of 8 kg. The answers were categorized as “Able to perform” and “Not able to perform”.

#### 2.2.7. Cognitive Function

In the present study, cognitive function was measured by the Korean version of the Mini Mental Status Examination for Dementia Screening (MMSE-DS) [27]. The MMSE-DS consists of 19 items, and the score ranges from 0 to 30. Lower scores indicate lower cognitive function.

### 2.3. Data Analysis

Data were analyzed using SPSS statistical software version 21.0 (IBM, Armonk, NY, USA). The general characteristics of participants and differences in variables between the DC group and the driving group were identified using the chi-square test or independent *t*-test. Univariate logistic regression analyses were used to identify the association between driving and DC. Only significant variables identified in the univariate logistic regression were entered in the multivariate logistic regression analysis (entered backward method). Multivariate logistic regression analyses were conducted to identify the predictors of DC. Correlation and multicollinearity between variables were confirmed by the tolerance and variance inflation factor (VIF). The predictors were estimated as an odds ratio (OR) and 95% confidence interval (CI). The statistical significance was set at the level of *p*-value < 0.05.

## 3. Results

### 3.1. General Characteristics of the Subjects and Differences in Variables between the DC Group and Driving Group

The general characteristics of the subjects and differences in the variables between the DC group and the driving group are summarized in Table 1. The age of all participants ranged from 65 to 91 years. The mean age of the participants who belonged to the DC group was 73.88 ± 6.13, and 1050 (35.4%) of 2970 had currently stopped driving.

### 3.2. Univariate Logistic Regression of DC

The results of univariate logistic regression of DC are shown in Table 2. In this study, six types of factors, including demographic, financial, psychosocial, environmental, physical, and cognitive factors, were classified based on the comprehensive mobility model. Most of the variables were significantly related to DC, except for education level (*p* = 0.147), visual discomfort (*p* = 0.974), and hearing discomfort (*p* = 0.637).

### 3.3. Multivariate Logistic Regression of DC

Multivariate logistic regression was performed to identify predictors of DC. Variables significantly associated with DC in the univariate analyses were entered into the regression model. The results are shown in Table 3 and Figure 1. Residential area, as an environmental factor, was a predictor of DC, with an odds ratio (OR) of 2.21 (95% confidential interval (CI) 1.86–2.62), indicating that subjects living in a city with a metro system were 2.21 more likely than subjects living in a city without a metro system to stop driving. In terms of demographic factors, older subjects (OR 1.14, 95% CI 1.12–1.16) and male subjects (OR 0.39, 95% CI 0.31–0.50) were observed to be a predictor of DC. In terms of financial factors, subjects with poor perceived financial status (OR 1.34, 95% CI 1.11–1.61) were more likely to stop driving than those with good perceived financial status. Regarding psychosocial factors, depression (OR 1.09, 95% CI 1.07–1.12) and a lack of participation in social activities (OR 1.66, 95% CI 1.38–2.00) were observed to be predictors of DC. Regarding physical factors, stroke (OR 1.51, 95% CI 1.09–2.09) and perceived difficulty climbing stairs (OR 1.56, 95% CI 1.26–1.93) were predictors of DC. In terms of cognitive factors, the MMSE-DS score (OR 0.92, 95% CI 0.89–0.96) was observed to be a predictor of DC.

## 4. Discussion

In this study, we determined that variables belonging to the demographic, financial, psychosocial, environmental, physical, and cognitive domains were predictors of DC based on the comprehensive mobility model. Participants’ average age was older in the DC group than in the driving group, and approximately 35% of participants aged 65 years and older had stopped driving in this study. In a study of 2932 Japanese elderly participants, 157 decided not to renew their driving license. Advanced age and female sex were determining factors of DC [28]. In a study of 5206 Australian elderly participants, the average of age was older in the non-driving group than the driving group, and 28.2% of males stopped driving [29]. Although the statistics were not directly compared with this study, like in the present study, these previous studies provide evidence that older drivers are concerned about driving.

In terms of environmental factors, a residential area with a metro systems was identified to be a strong predictor of DC. This result is consistent with previous studies [21,30]. Previous studies reported that living closer to public transit stops made older adults less likely to drive [30] and that older adults with limited access to alternative transportation were likely to use their personal vehicles [21]. In Japan, a high accessibility and convenience of public transportation impact on decision of DC has been reported among elderly adults [28]. The metro is a popular form of transportation for older adults in Korea. In particular, the free ride program for senior citizens aged 65 or older in Korea makes them likely to choose alternative transportation instead of personnel driving. In this study, older adults living in six residential areas with metros may have easily chosen to stop driving because of the high accessibility, convenience, and low cost of public transportation [31,32]. One study reported that older drivers in rural areas were driving longer than those in urban areas because of limited transportation support [33]. This study indicated that environmental factors could contribute to DC for older adults.

Regarding demographic factors, age and gender were predictors of DC. These variables were found to be predictors of DC in previous studies [34,35]. Older adults of advanced age were more likely to stop driving. Deterioration related to aging with respect to physical and cognitive function, such as reaction rate delay or cognitive decline [36], may have influenced the decision to stop driving. In terms of gender, female older adults were more likely to stop driving. This result reflects a greater difficulty for men compared with women to stop driving due to their role as the principle driver in households [33].

Subjects with poor perceived financial status were more likely to stop driving than those with good perceived financial status. Income and driving were previously found to be positively correlated [37], and fuel prices were negatively associated with driving [21]. In other words, high fuel prices may cause older adults to drive less [21], and the high cost of maintenance associated with driving may lead to DC in the elderly population [37].

Among psychosocial factors, depression and social activity were identified as predictors of DC. A previous study that aimed to confirm the significant factors associated with DC in older Malaysian adults reported that depression was significantly associated with DC [38]. Another study reported that depression was related to social connections and that elderly people with depression who had fewer social connections were likely to stop driving. Furthermore, depression exacerbated by DC could cause self-restricted mobility and social isolation and could ultimately affect the mobility and quality of life of older adults [1,39,40]. Therefore, it should be considered that efforts for DC that do not provide suggestions regarding alternative transportation or the improvement of mobility after DC could lead to other issues, such as social isolation and decreased quality of life among older adults.

Among physical factors, stroke and lower extremity function were identified as predictors of DC. Stroke, as a neurologic problem, leads to decreases in visual field, attention and information processing speed and physical function, eventually causing stroke survivors to reduce or cease driving [41,42]. The results showed that older adults who had difficulty climbing stairs as an indicator of lower extremity function were 1.50 times more likely to stop driving. Changes in muscle strength and flexibility affect changes in muscle length of the lower extremity musculature [43] and lead to a decrease in lower extremity function, eventually increasing the DC rate [44]. Therefore, the assessment of physical function, such as lower extremity function, is required to determine the need for DC. However, in terms of stroke survivors, one study reported that 66.1% of patients with stroke returned to driving within one year after the stroke [42]. Another study reported that a continuous exercise rehabilitation program allowed stroke patients to recover their extremity function and driving ability [45]. Therefore, based on a screening test, if older adults have physical function that is suitable for driving or if their physical function can be improved through rehabilitation, information and education on rehabilitation should be provided to maintain or increase their mobility. Periodic assessments of physical function and medical check-ups should be performed for people who have the possibility to recover their physical function through rehabilitation. However, if it is not possible for older adults to increase their function or achieve proper function to drive, DC is strongly suggested for public safety. Based on this result, physical factors such as stroke and lower extremity function were identified as predictors of DC. Although the current study did not identify vision or hearing discomfort as predictors of DC, other studies have indicated that these sensory functions are very important for driving [17,34,41]. Therefore, further study to identify the influence of sensory function is required.

Visual and hearing discomfort were excluded from the predictor in univariate and multivariate analysis. This result is inconsistent with a previous study showing that vision is a potentially important predictor of DC [35]. The present study represents a second data analysis, and vision and hearing functions as predictors of DC were only measured by subjective conditions, such as visual and hearing discomfort. Therefore, we suggest that further study should include objective or quantitative measurements of these variables.

The results show that with every one-point increase in the cognitive function score, the likelihood of DC decreased by 0.92 times. This result is consistent with a previous study showing that cognitive impairment was a strong predictor of DC [34]. In this study, cognitive function was evaluated by the MMSE-DS score, but several studies have reported that evaluation by the MMSE score has limitations for predicting DC [12,46]. Kosuge and colleagues suggested that a reduction in cognitive processing speed is the strongest predictor of DC rather than the MMSE score [12]. Another study reported that flanker task scores, selective attention, and inhibitory ability are stronger DC predictors than the MMSE score [46]. Regardless of the measurement used to evaluate cognition, cognitive function was identified as an important predictor of DC based on the results and previous studies. Therefore, it is necessary to assess or manage cognitive function related to driving.

In this study, each factor in the comprehensive mobility framework [1] was identified as a predictor of DC. Driving seems to be a simple skill, but it requires various and complex functions and abilities. Therefore, comprehensive demographic, financial, psychosocial, environmental, physical, and cognitive factors should be considered in the recommendation of DC for high-risk drivers. In addition, when DC is suggested for high-risk drivers, adverse consequences associated with DC should be identified, and multidisciplinary approaches to creating an age-friendly environment will be required to accommodate all older adults who continue driving or stop driving.

There are several limitations of this study because it is a second data analysis using the 2017 National Survey of Elderly People. First, this study did not include various variables related to driving ability, such as cognitive process factors (e.g., risk perception, executive function, working memory, attention, and speed of information processing) [36]. Second, this study included subjective variables, such as financial status or sensory problems, because driving requires not only various objective function but also subjective driving self-confidence [47]. Therefore, further studies should include various subjective and objective variables to predict DC in the older population.

## 5. Conclusions

This study identified the predictors of DC among older adults based on a comprehensive mobility model. Demographic, financial, psychosocial, environmental, physical, and cognitive variables of the comprehensive mobility model were identified as predictors of DC. Therefore, female, older drivers with an advanced age, poor financial status, decline of psychosocial and physical function, and lower MMSE-DS score were likely to stop driving. In particular, a residential area with highly accessible public transportation, as an environmental factor, was a strong predictor of DC. Older adults who were living in an area with a metro system showed a high DC rate. In terms of the mobility of older adults, maintaining mobility by driving is closely related to quality of life as well as maintaining independence. Therefore, political or environmental support is essential to increase DC to reduce traffic accidents involving high-risk older drivers without decreasing their mobility. In addition, it is necessary to develop safe driving guidelines for older drivers, including a screening test for cognitive function, taking a driving lesson, and an objective test of driving ability before renewing their driver’s licenses. Factors identified as DC predictors in this study and additional factors, such as objective financial factors, sensory factors, or cognitive process factors (e.g., risk perception, executive function, working memory, attention, and speed of information processing), should be considered in the determination of whether older adults should continue or stop driving.

## Figures and Tables

**Figure 1 ijerph-17-07206-f001:**
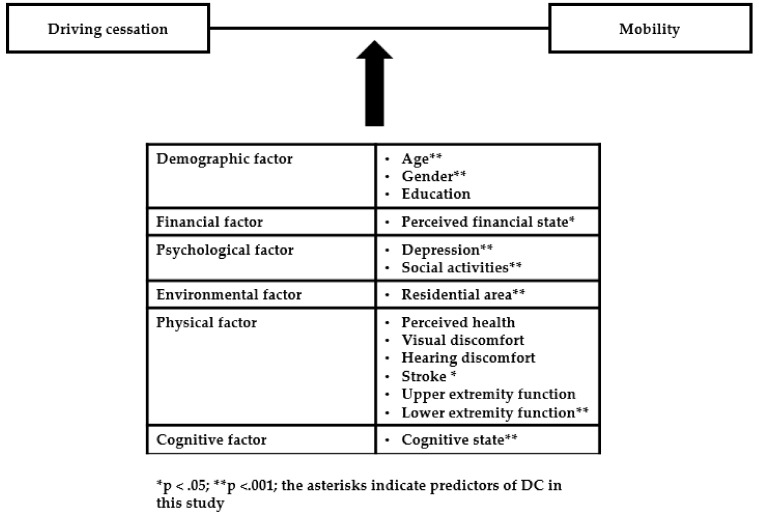
Factors of driving cessation based on a comprehensive mobility framework in this study.

**Table 1 ijerph-17-07206-t001:** General characteristics of the subjects and differences in variables between the driving cessation group and driving group (*N* = 2970).

Characteristics	Variable	Driving (*n* = 1920, 64.6%)	Driving Cessation (*n* = 1050, 35.4%)	*t* or x^2^ (*p*)
		*n* (%) or Mean ± SD	*n* (%) or Mean ± SD	
Demographic	Age (year)	70.23 ± 4.45	73.88 ± 6.13	−17.02 (<0.001)
factor	Gender			
	Female	244 (12.7)	200 (19.0)	21.35 (<0.001)
	Male	1676 (87.3)	850 (81.0)	
	Education (year)	4.57 ± 1.39	4.48 ± 1.41	1.66 (0.097)
Financial	Perceived financial state			
factor	Good	859 (44.7)	332 (31.6)	8.91 (<0.001)
	Poor	1061 (55.3)	718 (68.4)	
Psychological	Depression (score)	2.22 ± 3.08	4.25 ± 4.32	−13.47 (<0.001)
factor	Social activities			
	Yes	1414 (73.6)	540 (51.4)	149.57 (<0.001)
	No	506 (26.4)	510 (48.6)	
Environmental	Residential area			
factor	Living in city with metro system	664 (34.6)	540 (51.4)	79.71 (<0.001)
	Living in city without metro system	1256 (65.4)	510 (48.6)	
Physical	Perceived health			
factor	Good	1123 (58.5)	415 (39.5)	97.79 (<0.001)
	Poor	797 (41.5)	635 (60.5)	
	Visual discomfort			
	Yes	665 (34.6)	358 (34.1)	0.10 (0.753)
	No	1255 (65.4)	692 (65.9)	
	Hearing discomfort			
	Yes	332 (17.3)	189 (18.0)	0.22 (0.650)
	No	1588 (82.7)	861 (82.0)	
	Stroke			
	Yes	101 (5.3)	116 (11.0)	33.48 (<0.001)
	No	1819 (94.7)	934 (89.0)	
	Lower extremity function			
	Sit and stand up			
	Able to perform	1836 (95.6)	883 (84.1)	116.63 (<0.001)
	Not able to perform	84 (4.4)	167 (15.9)	
	Climbing stairs			
	Able to perform	1651 (86.0)	685 (65.3)	173.69 (<0.001)
	Not able to perform	269 (14.0)	365 (34.7)	
	Upper extremity function			
	Lifting or moving objects (8 kg)			
	Able to perform	1796 (93.5)	859 (81.8)	98.54 (<0.001)
	Not able to perform	124 (6.5)	191 (18.2)	
Cognitive factor	MMSE-DS (score)	27.37 ± 2.30	26.44 ± 2.97	8.83 (<0.001)

SD: standard deviation; MMSE-DS: Mini Mental Status Examination for Dementia Screening.

**Table 2 ijerph-17-07206-t002:** Univariate logistic regression models for driving cessation (*N* = 2970).

Characteristics	Variable	OR	95% CI		*p*
			Upper	Lower	
Demographic factor	Age (year)	1.14	1.12	1.15	<0.001
	Gender				
	Female	0.62	0.51	0.76	<0.001
	Male (ref.)				
	Education (year)	0.99	0.97	1.01	0.147
Financial factor	Perceived financial status				
	Good (ref.)				
	Poor	1.75	1.50	2.05	<0.001
Psychosocial factor	Depression	1.16	1.13	1.18	<0.001
	Social activities				
	Yes (ref.)				
	No	2.64	2.26	3.09	<0.001
Environmental	Residential area				
factor	With metro system	2.04	1.72	2.34	<.001
	Without metro system (ref.)				
Physical factor	Perceived health				
	Good (ref.)				
	Poor	2.16	1.85	2.52	<0.001
	Visual discomfort				
	Yes	0.97	0.83	1.14	0.974
	No (ref.)				
	Hearing discomfort				
	Yes	1.05	0.86	1.28	0.637
	No (ref.)				
	Stroke				
	Yes	2.23	1.69	2.95	<0.001
	No (ref.)				
	Lower extremity function				
	Sit to stand				
	Able to perform (ref.)				
	Not able to perform	4.14	3.15	5.45	<0.001
	Climbing stairs				
	Able to perform (ref.)				
	Not able to perform	3.26	2.72	3.91	<0.001
	Upper extremity function				
	Lifting or moving objects (8 kg)				
	Able to perform (ref.)				
	Not able to perform	3.23	2.54	4.11	<0.001
Cognitive factor	MMSE-DS (score)	0.87	0.85	0.90	<0.001

ref: reference; OR: odds ratio: CI: confidential interval; MMSE-DS: Mini Mental Status Examination for Dementia Screening.

**Table 3 ijerph-17-07206-t003:** Multivariate logistic regression models for driving cessation (*N* = 2970).

Characteristics	Variable	OR	95% CI		*p*
			Upper	Lower	
Demographic factor	Age	1.14	1.12	1.16	<0.001
	Gender				
	Female (ref.)				
	Male	0.39	0.31	0.50	<0.001
Financial factor	Perceived financial state				
	Good (ref.)				
	Poor	1.34	1.11	1.61	0.003
Psychological factor	Depression	1.09	1.07	1.12	<0.001
	Social activities				
	Yes (ref.)				
	No	1.66	1.38	2.00	<0.001
Environmental factor	Residential area				
	With metro system	2.21	1.86	2.62	<0.001
	Without metro system (ref.)				
Physical factor	Stroke				
	Yes	1.51	1.09	2.09	0.012
	No (ref.)				
	Lower extremity function				
	Climbing stairs				
	Able to perform (ref.)				
	Not able to perform	1.56	1.26	1.93	<0.001
Cognitive factor	MMSE-DS (score)	0.92	0.89	0.96	<0.001
Constant		<.001			<0.001
Hosmer–Lemeshow test: x^2^ = 11.23, *df* = 8, *p* = 0.189 Model summary: Nagelkerke R^2^ = 0.293					

ref: reference; OR: odds ratio; CI: confidential interval; MMSE-DS: Mini Mental Status Examination for Dementia Screening.

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
