# Peer review of "The Predictors of Driving Cessation among Older Drivers in Korea"

_ijerph, 2020, doi:10.3390/ijerph17197206_

Round 1
Reviewer 1 Report
A study aimed to explore the driving cessation (DC) rate among older Korean adults and predictors of DC based on a comprehensive mobility model (cognitive, psychosocial, physical, environmental, and financial factors) is carried out in this paper and it is the main subject the authors deal with. Such model analyzed data from 2,970 of 10,062 older adults over 65 years old from the National Survey of Elderly People (2017). Based on multivariate logistic regression analyses, authors argue residential area with a metro system (an environmental factor), was a strong predictor of DC. However, in general, environmental factors were strong predictors of older adults’ DC. In this sense, authors propose that results of this study could provide a guideline in Korea about political and environmental support (e.g., such as the provision of accessible public transportation) to increase the DC rate among older adults, and to increase public safety without decreasing their mobility. However, I think that the authors should make an effort to improve the paper by taking into account the following remarks:
- A comparison of the results with other countries with similar characteristics to those of Korea (tables, metrics, etc.) is desirable to firmly corroborate the decisions that could be derived from this study.
- It is desirable to include some figures and pictures (with more and better-detailed information about) that complement and graphically improve the explanations and results given in this manuscript.
- Conclusions are too general and they are no consequence of a rigorous results analysis. In this sense, the recommendations related to a more robust public transportation because older adults who were living in an area with a metro system showed a high DC rate, as well as political or environmental support to increase DC and reduce traffic accidents involving high-risk older drivers without decreasing their mobility, seem to be common-sense conclusions, that need not be supported by the results obtained. Is it necessary to do this study to affirm these conclusions? The authors should give more explanation about it.
- Future work is not appropriately addressed in the paper. “it is necessary to develop safe driving guidelines for older drivers” which kind of guidelines? and “various factors affecting driving should be considered in the determination of whether older adults should continue or stop driving”, examples of such additional factors?
Author Response
- 1. A comparison of the results with other countries with similar characteristics to those of Korea (tables, metrics, etc.) is desirable to firmly corroborate the decisions that could be derived from this study.
- Answer: Thank you for the suggestion. It is difficult to find the data from other countries with similar characteristics to those of Korea. We have attempted to add the information of Japan and Australia. Even though, characteristics of these countries are a little different from our county, we think that these information will be helpful to support our results (Line 205~212).
- 2. It is desirable to include some figures and pictures (with more and better-detailed information about) that complement and graphically improve the explanations and results given in this manuscript.
- Answer: Thank you for your comments. On line 107, we have added the conceptual framework to provide readers with a better understanding (Figure 1).
- 3.Conclusions are too general and they are no consequence of a rigorous results analysis. In this sense, the recommendations related to a more robust public transportation because older adults who were living in an area with a metro system showed a high DC rate, as well as political or environmental support to increase DC and reduce traffic accidents involving high-risk older drivers without decreasing their mobility, seem to be common-sense conclusions, that need not be supported by the results obtained. Is it necessary to do this study to affirm these conclusions? The authors should give more explanation about it
- Answer: Thank you for your comments. As your opinion, political or environmental support to increase DC and reduce traffic accidents involving high-risk older drivers without decreasing their mobility seem to be common-sense conclusions. I partially agree with you. However, we think that there are not enough studies suggested how to obtain the both objectives such as increasing DC involving high-risk older drivers and maintaining their mobility. Our main suggestion is the balance between the mobility of the elderly and public safety. We have revised Introduction, and Conclusion sections to help readers better understand our suggestion (lines 56-71, 313-326).
- 4. Future work is not appropriately addressed in the paper. “it is necessary to develop safe driving guidelines for older drivers” which kind of guidelines? and “various factors affecting driving should be considered in the determination of whether older adults should continue or stop driving”, examples of such additional factors?
-
Answer: We have added a sentence about future work on lines 320~326.
Reviewer 2 Report
The author studied 2970 people over 65 years old in South Korea. Multivariate logistic regression analyses were conducted to identify the predictors of DC. The results show that residential area, demographic, financial, psychosocial, physical, and cognitive variables are predictors of DC. Therefore, political or environmental support is essential to increase DC to reduce traffic accidents without decreasing their mobility.
The article has a complete structure and clear thinking, which has a certain value of in-depth research. But there are still some problems in the article as follows.
1.The maximum age of participants is not stated.
2.The logic of the second paragraph of the introduction is unreasonable, and the author is suggested to improve it.
3.It only analyzes the results of multiple logistic regression analysis, but the suggestions made are not specific enough and targeted.
4.The reason for the possibility of insignificant predictors (education level, visual or auditory discomfort) is not explained.
5.Except for physical factors and cognitive functions, the quantitative relationship between several other predictive factors and stopping driving is not stated.
6.The variables considered in the article are not complete. Other variables that may be related to stopping driving, such as risk perception and driving skills, can be added in the discussion.
7.The conclusion part shall elaborate and summarize the effect of each predictive factor.
Author Response
- The maximum age of participants is not stated.
Answer: The age of all participants ranged from 65 to 91 years. We have added the age range on line 173.
- The logic of the second paragraph of the introduction is unreasonable, and the author is suggested to improve it.
Answer: To clarify the explanation in the paragraph, we have revised and rearranged the text (lines 56~71).
- It only analyzes the results of multiple logistic regression analysis, but the suggestions made are not specific enough and targeted.
Answer: Thank you for your comments. The results of multiple logistic regression analysis were that demographic, financial, psychosocial, environmental, physical, and cognitive domains were predictors of DC. We have attempted to revise and reorganize Discussion and Conclusion sections to be specific and targeted with additional references, and we have also included information related to other reviewers’ comments in these sections, which are highlighted in red.
- The reason for the possibility of insignificant predictors (education level, visual or auditory discomfort) is not explained.\
Answer: We have added the description about insignificant predictors on line 277-282.
- Except for physical factors and cognitive functions, the quantitative relationship between several other predictive factors and stopping driving is not stated.
Answer: We have added information about other predictors such as demographic factors with a relationship to DC and reorganized other factors (financial, psychosocial, environmental, physical, and cognitive factors) related to DC (Line 205-212, 276-282).
- The variables considered in the article are not complete. Other variables that may be related to stopping driving, such as risk perception and driving skills, can be added in the discussion.
Answer: We agree that risk perception and driving skills are very important predictors of DC. However, this study was a second data analysis and was not equipped to analyze other variables not included in the data set (the 2017 National Survey of Elderly People). This issue has been described in the Conclusion section and suggestions provided for further study (line 300-303).
- The conclusion part shall elaborate and summarize the effect of each predictive factor.
Answer: Thank you for your comments. We have added some text in the Conclusion section to be elaborate and summarize the effect of each predictive factor (line 311-313).

Reviewer 3 Report
This study tried to identify the predictors of driving cessation (DC) rate of older drivers and give suggestions for how to reduce the DC rate of older drivers without sacrifice their mobility. The author has done some researches but the basic argument that older drivers are tend to harm the public safety remained to be discussed. It is a form of discrimination to generalize all elderly drivers. Here are some suggestions for the authors to consider for further revision.
- Line64,65: ‘and another study indicated that older drivers…to driving.’ This statement seems show the opposite view towards the study. Some explanation is suggested.
- Line116: There are only two options for subjects, which may not suitable for people who only drive occasionally. Detailed classification is suggested. Or there need to be a clear definition for DC, which explain that even once driving is not allowed.
- Line123: For financial factor, there are only subjective options good or poor. Objective data were suggested for further analysis. The article also discussed the pension issue in Discussion, but there is no data to support it. Only a subjective opinion about whether he/she is poor is not enough
- Line137: How to evaluate the visual and hearing discomfort should be instructed.
- Line198: The conclusion in this paper seems to have little to do with references 29 and 30. It only found that a residential area with a Metro systems was a strong predictor of DC, but it did not mention the relationship of density and accident rate. Sufficient support is needed.
- Line277: there is some confusion of the statement ‘DC can result in poor outcomes of psychosocial and cognitive function for older adults.’. The logistics model can only verify correlation and conclusion may be the DC result from poor outcomes of psychosocial and cognitive function for older adults. Or there need to be more argument to support this view.
Author Response
- Line64,65: ‘and another study indicated that older drivers…to driving.’ This statement seems show the opposite view towards the study. Some explanation is suggested.
- Answer: Yes, we have revised the paragraph to clarify the description (line 56-71).
-
Line116: There are only two options for subjects, which may not suitable for people who only drive occasionally. Detailed classification is suggested. Or there need to be a clear definition for DC, which explain that even once driving is not allowed.
-
Answer: Thank you for your comments. This study has limitations because it is a second data analysis, so we can`t classify whether the older drivers drive usually or occasionally. We have provided a definition for DC to clarify the classification of drivers in the measurement section (line 121-126).
- Line123: For financial factor, there are only subjective options good or poor. Objective data were suggested for further analysis. Only a subjective opinion about whether he/she is poor is not enough
- Answer: Thank you for your opinion. This study is a second data analysis, so we can`t measure the objective financial condition. We think that driving requires not only various objective conditions or functions, but also subjective condition such as driving self-confidence. Therefore, we think that subjective variables could also enter the regression model to identify predictors of DC. However, I agree with your opinion regarding the pension issue is not suitable to support it, so we deleted the text (line 243). We have also added limitation about including subjective variables in the Discussion section (line 303-305).
- Line137: How to evaluate the visual and hearing discomfort should be instructed.
- Answer: We have added some text concerning how to evaluate visual and hearing discomfort (Line 145-148).
-
Line198: The conclusion in this paper seems to have little to do with references 29 and 30. It only found that a residential area with a Metro systems was a strong predictor of DC, but it did not mention the relationship of density and accident rate. Sufficient support is needed.
Answer: Thank you for your comments. We have deleted the sentence related to road density or traffic congestion (line 214-221). Additionally, we have provided a more detailed explanation for the relationship between the transportation system and driving (221-232).
- Line277: there is some confusion of the statement ‘DC can result in poor outcomes of psychosocial and cognitive function for older adults’. The logistics model can only verify correlation and conclusion may be the DC result from poor outcomes of psychosocial and cognitive function for older adults. Or there need to be more argument to support this view.
- Answer: We agree with the reviewer and have deleted the sentence for clarity (line 317).

Round 2
Reviewer 1 Report
The authors addressed appropriately to the comments and suggestions made. However, Figure 1 is not useful. It does not show better and detailed information that complements and graphically improve the explanations and results given in this manuscript.
Author Response
- The authors addressed appropriately to the comments and suggestions made. However, Figure 1 is not useful. It does not show better and detailed information that complements and graphically improve the explanations and results given in this manuscript.
Answer : Thank you for your comments. On line 203, we have revised the Figure 1 including results given in this manuscript. We have moved the place of the figure 1 and changed the title of the figure 1.

Reviewer 2 Report
This paper has been greatly improved in all aspects after the author's revision. It can be found that the discourse of the paper is clearer, the logic is more rigorous, the structure is more complete, and the use of language has been greatly improved. In addition, I also found a problem in my review process for the author's reference:
The logic of the second and third paragraphs needs to be improved. It is recommended that the author first explain the definition of mobility and its importance to the elderly, and then state whether elderly driving has a relationship with the accident rate.
Author Response
Answer : Thank you for your comments. We have revised the order of paragraphs according to your recommendation (Line 30~40). We explained the definition of mobility firstly and stated the explanation about car accidents by older drives.
